# Tracking And Understanding Public Reaction During COVID-19: Saudi Arabia As A Use Case

**Aseel Addawood**
Information Systems Department
Imam Mohammad Ibn Saud Islamic Uni.
aasdawood@imamu.edu.sa

**Alhanouf Alsuwailem**
Information Systems Department
Imam Mohammad Ibn Saud Islamic Uni.
aasalsuwailem@imamu.edu.sa

**Ali Alohali**
Bayan foundation
ali@alioh.com

**Dalal Alajaji**
Bayan foundation
dalal.a.aj@gmail.com

**Mashail Alturki**
Bayan foundation
mashyt55@gmail.com

**Jaida Alsuhaibani**
Saudi Technology and Security
Comprehensive Control Company
Jalsuhaibani@tahakom.com

**Fawziah Aljabli**
Bayan foundation
Fawziah.aljabli@outlook.sa

## Abstract

The coronavirus disease of 2019 (COVID-19) has a huge impact on economies and societies around the world. While governments are taking extreme measures to reduce the spread of the virus, people are getting affected by these new measures. With restrictions like lockdown and social distancing, it became important to understand the emotional response of the public towards the pandemic. In this paper, we study the reaction of Saudi Arabia citizens towards the pandemic. We utilize a collection of Arabic tweets that were sent during 2020, primarily through hashtags that were originated from Saudi Arabia. Our results showed that people had kept a positive reaction towards the pandemic. This positive reaction was at its highest at the beginning of the COVID-19 crisis and started to decline as time passes. Overall, the results showed that people were so supportive of each other through this pandemic. This research can help researchers and policymakers in understanding the emotional effect of a pandemic on societies.

## 1 Introduction

COVID-19 has been declared as a pandemic by the World Health Organization (WHO) on January 30, 2020 (Organization et al., 2020). The spread of COVID-19 around the world has raised many concerns and uncertainties about the upcoming future. Several epidemic periods have been observed in the world. In recent years, this epidemic has grown because of the contagion favored by globalization. The pandemic has affected 213 countries, with more than one and a half million confirmed cases to date (Organization et al., 2020).

To overcome the rapid spread of the virus, the World Health Organization (WHO) has suggested that isolation and self-quarantine is one of the major ways to stop this pandemic to spread at such an alarming rate (Cucinotta and Vanelli, 2020). China has witnessed the benefits of one of the largest lockdowns at the start of this pandemic, where it locked down 20 provinces and regions (Koh, 2020).

One of the most widely used social media platforms is Twitter, popular for its accessibility and ease of information sharing (Wang et al., 2016). With the increased precautionary measures used by governments such as lockdown and curfew, people shift to social media platforms to continue their social interactions with a 61% increase in the usage of these platform (Holmes, 2020). With 13.8 million active users out of 24 million internet users (91% of the total population), Saudi Arabia is among the countries with the highest number of Twitter users among its online population (Clement, 2020; Puri-Mirza, 2019). Moreover, Saudi Arabia is producing 40% of all tweets in the Arab world (Mourtada and Salem, 2014).

Since a majority of online communication is recorded in the form of text data, measuring the emotions around COVID-19 will be a central part

of understanding and addressing the impacts of the COVID-19 situation on people. Moreover, the rapid spread of COVID-19 infections has created a strong need for understanding the development of mass sentiment in pandemic scenarios. The uncertainty that is provoked by the COVID-19 situation and the many and long-lasting preventative measures applied, it is of vital importance to understand how governments, NGOs, and social organizations can help those who are most affected by the situation and where they are located. To be able to do so, understanding the emotions, worries, and concerns that people have and possible coping strategies is important.

Previous research in sentiment analysis used Twitter platform to collect tweets in various languages such as: Indian , Nepal, Turkish ,Arabic, and Spanish to detect the tweets' sentiments during a crisis (Bhat et al., 2020; García-Díaz et al., 2020; Barkur and Vibha, 2020; Pokharel, 2020; Manguri et al., 2020; Baker et al., 2020; Alhajji et al., 2020; Öztürk and Ayvaz, 2018; Adel and Wang, 2019; Aloqaily et al., 2020). Recently, Arabic language got more attention in sentiment analysis since it is becoming widely used in social media platforms (Aloqaily et al., 2020). As yet, many research on Arabic sentiment analysis conducted to detect opinions about crisis and epidemic as well (Baker et al., 2020; Alhajji et al., 2020; Adel and Wang, 2019; Aloqaily et al., 2020).

In this study, we try to understand Saudis' reactions towards this pandemic, by answering three questions:
- What is the overall Saudi citizens' sentiment to the pandemic?
- What is the geographical distribution of different sentiments?
- How Saudis reacted toward different topics/decisions related to the pandemic?

Analyzing people's emotions regarding COVID-19 crisis's restrictions will be a central part of understanding and addressing the impacts of pandemics on people. Moreover, this study might inform the development of models of social behavior that fit the Saudi Arabic culture. Our results showed a high solidarity between the society, where people were expressing positive sentiment in hashtags related to social solidarity. Also, our results showed there was a high correlation between the number of COVID-19 cases and the geolocation of the tweets.

In the remainder of the paper, we first synthesize previous work and background information and relate that to our work. After that, we describe the data set used for the analysis. We then discuss sentiment identification. Finally, we discuss the findings, conclusions, and proposed directions for future research.

## 2 Related Work

### 2.1 Sentiment Analysis

There is a technique related to natural language processing (NLP) known as sentiment analysis that helps detect and extract the polarity of sentiments from text by determining if the text is positive, negative, or neutral (García-Díaz et al., 2020). There are two approaches to sentiment analysis: machine learning and lexicon-based. The machine learning approach uses algorithms to find sentiment, while the lexicon-based approach counts positive and negative words (Drus and Khalid, 2019).

Social media is widely used by the public to express opinions, ideas, and emotions, thus allowing researchers to analyze users' sentiments (Bhat et al., 2020). In Drus and Khalid (2019), researchers divided social media platforms into four categories depending on their use: content communities such as YouTube and Instagram, social networking sites such as Facebook and LinkedIn, blogs, and microblogs as Twitter and Tumblr. Of these types, the top category for collecting information was microblogs, especially Twitter, as it is considered as one of the most popular websites and allows users to post only short messages (Öztürk and Ayvaz, 2018); Twitter limits users by providing them with only 280 characters for each tweet (Pokharel, 2020). Also, microblogging attracts researchers focusing on various languages. Twitter publishes around 500 million daily tweets; in comparing the number of active Twitter users and overall internet users in the world, Twitter users number more than 22% of the overall internet population (Öztürk and Ayvaz, 2018). There were two other main reasons for choosing Twitter in García-Díaz et al. (2020): (1) It is a popular platform for spreading news and information, being established as an appropriate means to collect information about public health; and (2) the hashtags common to tweets help the social network act as a hub-and-spoke.

To highlight the major role of sentiment analysis on a crisis,the research conducted by Öztürk and Ayvaz (2018) aims to explore sentiments about civil war in Syria as well as the refugee crisis, using

Twitter data consisting of 2,381,297 tweets in Turkish and English. The reason for selecting Turkish was the huge number of Syrian refugees based in Turkey. The final result demonstrates that Turkish tweets expressed positive sentiments about Syrians and refugees at a rate of 35%, while English tweets contained 12% positive sentiments.

Furthermore, in García-Díaz et al. (2020), an aspect-based sentiment analysis is used to define the tweets' sentiments by using deep-learning models with other techniques, such as word-embedding and linguistic features, on Spanish corpus about infectious diseases such as Dengue, Zika, or Chikungunya viruses in Latin America. The corpus includes 10,843 positive tweets, 10,843 negative tweets and 7,659 neutral tweets. The researcher of Adel and Wang (2019) provided the first Arabic corpus for crisis-response messages as there is no Arabic corpus associated with humanitarian crisis since the most humanitarian crisis is in the Arab countries. Thus, this corpus created based on Twitter data and it includes tweets about cholera epidemic and starvation in Yemen and the Syrian refugees. The process of creating the corpus followed six steps: (1) Collecting data (2) Annotation process of Arabic tweets (3) Prepossessing of text (4) Feature extraction (5) Model building and (6) Model evaluation. Furthermore, the supervised machine learning model used to predict the label for Arabic tweets, three experiments were conducted by using SVM, NB, and Random Forest (RF), the best classifier result revealed by RF.

Aloqaily et al. (2020) used two main approaches of sentiment analysis: lexicon-based and machine learning on Arabic tweets about civil war in Syria and crises. The results demonstrated that machine learning performs better than the lexicon-based. Regarding machine learning, five algorithms were used: the Logistic Model Trees (LMT), simple logistic, SVM, DT, Voting- based and k-Nearest Neighbor (K-NN) , the best performance result performed by LMT with Accuracy = 85.55, F1= 0.92 and Area Under Curve (AUC)= 0.86.

## 2.2 Sentiment Analysis during an epidemic

In a sentiment analysis of 615 tweets from Nepal, using the hashtags COVID-19 and coronavirus, the final result showed that about 58% of users' tweets were optimistic, 15% were negative tweets, and 27% were neutral (Pokharel, 2020). On the other hand, Barkur and Vibha (2020) presented a sentiment analysis of Indians' tweets about the lockdown by collecting data from Twitter using two hashtags: #IndiaLockdown and #IndiafightsCorona. This resulted in 24,000 tweets, dominated by positive sentiments. In another study, researchers collected five hundred thirty thousand by using the keywords coronavirus and COVID-19. More than 36% of users posted positive tweets, while the negative tweets numbered only 14%, and the neutral tweets accounted 50% (Manguri et al., 2020). Further, Bhat et al. (2020) used two hashtags to perform sentiment analysis related to the current coronavirus pandemic: #COVID-19 resulted in 92,646 collected tweets, and #Coronavirus resulted in 85,513 collected tweets. Regarding the COVID-19 hashtag, most tweets had positive sentiments (51.97%). Neutral sentiments represented 34.05% of tweets, and 13.96% relayed negative sentiments. On the other hand, sentiments related to Coronavirus were more neutral (41.27%). Positive tweets accounted for 40.91% of the overall numbers and 17.80% for negative sentiments. Overall, the results indicate that users' opinions were mostly positive or neutral.

## 2.3 Arabic Sentiment Analysis during epidemic

The authors of (Baker et al., 2020) suggested a sentiment analysis system based on 54,065 Arabic tweets related to influenza and labeling the data as valid or invalid based on their relation to influenza. Also,they used different machine learning techniques such as support vector machine (SVM), naïve Bayes (NB), K-nearest neighbor (k-NN), and decision tree (DT). The outcomes of three experiments show that the NB algorithm has the best value, with a 20-fold accuracy of 83.20%. From Arab countries, Syria provides the highest value of 89.06% with the NB algorithm, while the Arabian Gulf region achieved the greatest accuracy of 86.43% with k-NN. SVM performed poorly in all three experiments.

Recently, in research conducted by Alhajji et al. (2020) to study the attitude of Saudis related to COVID-19 preventive processes by applying Arabic sentiment analysis using Twitter posts, the NB algorithm was applied to 53,127 tweets collected based on seven health measures: - Great mosque closures - Qatif lockdown - Schools and university closures - Public park, restaurant, and mall closures - Suspension of all sports activities - Suspension

of Friday prayers - Curfew for 21 days The final conclusion of this study is that Saudi Twitter users are willing to support and have positive opinions of health measures announced by the Saudi government. Religious practices also play an important role by increasing the overall positive sentiment.

## 3 Data

Our primary dataset is a repository of Tweet IDs each corresponds to Arabic content posted on Twitter and related to the Coronavirus pandemic (Addawood, 2020). The repository contains 3.8 million Tweet IDs for the period of January 1, 2020, to April 10, 2020. Tweets were compiled utilizing *Crimson Hexagon*, which is a social media analytic platform that provides paid data stream access. The data was collected by identifying a list of trending hashtags and keywords mostly used by the public. The hashtags were categorized based on their meaning, how users were interacting with it and who started the hashtags. The categorization was done by Addawood (2020). Table 1 shows a sample of the hashtags.

For collecting a comprehensive dataset, 70 keywords, and hashtags were categorized based on how they were oriented and used in conversations. Table 2 shows the hashtag topics with the number of tweets for each topic. The data was divided into four time periods as shown in table 3 to understand the public opinions as the pandemic evolves.

## 4 Method

### 4.1 Sentiment Lexicon

Before working on identifying the sentiment, the dataset needs to be cleaned. The cleaning process included removing username, punctuation, numeric characters, links, English characters, stop words, and normalizing the Arabic text. After measuring the average length of characters before and after the cleaning process, we found that the average length of tweets before cleaning was 184.63 characters and after cleaning it is 142.23 characters. Sentiment analysis techniques allowed researchers and business people to determine the different viewpoints expressed in social media text. The availability of a comprehensive Arabic sentiment lexicon is limited, for that, we built our own lexicon. The lexicon was built using two methods. The first method was by using previously constructed lexicons for Arabic sentiment (Al-Thubaity et al., 2018; Al-Twairesh et al., 2016; Salameh et al., 2015).

These lexicons either did not contain words specifically suited for Saudi dialect or they where domain specific. Our goal was to be able construct a Saudi dialect sentiment lexicon that can generalize to other domains. The second method used to build the lexicon was by utilizing different textual data sources in Arabic such as Twitter data (Nabil et al., 2015), book review (Aly and Atiya, 2013), hotel reviews (Elnagar et al., 2018). For each of these data sets, the annotation process of posts was done by the first two authors which they manually labelled each textual data point and extracted words expressing emotions, these data sets were divided between them and each data set was annotated by one annotator. The authors are currently working on developing a comprehensive Saudi dialect lexicon with having 9 annotators. The resulted lexicon contained 7,534 words, 1,808 (24%) of it was labeled as positive and 5,726 (76%) was labeled as negative.

### 4.2 Identification of Author Location

The globalization of the pandemic allowed people from all over the world to participate in this discussion. To find out where Twitter users were located, the location was available in the data set and it was retrieved as provided by *Crimson Hexagon*. To infer locations, Crimson Hexagon uses two types of information: 1) geotagged locations, which are only available for about 1% of Twitter data (Jurgens et al., 2015; Morstatter et al., 2013); and 2) for tweets that are not geotagged, an estimation of the users' countries, regions, and cities based on "various pieces of contextual information, for example, their profile information", as well as users' time zones and languages.

### 4.3 Sentiment Identification

Our goal is to have a full understanding of the public reactions towards the pandemic in Saudi Arabia. To be able to identify the sentiment of each tweet in our dataset, we used a machine learning approach. To do so and since the data set is huge, we selected a sample of the data set to annotate based on the created lexicon. The sample was chosen based on selecting tweets with a text length between 160 and 170 characters, the average length of all tweets was 184.63 characters before cleaning and 142.23 characters after cleaning. After reviewing the whole data, we decided to take a small sample of it for sentiment identification because we will use that sample to train the machine learning models. In-

| Hashtag | English Translation | Hashtag type / Populated by |
|---|---|---|
| كلنا _مسؤول | We are all responsible | Saudi governmental |
| الوقاية _من _كورونا | Corona Prevent | accounts |
| إيقاف _صلاة _الجماعة | Stopping congregational prayer | Discussing precautionary measures applied by the |
| تعليق _الدراسة | School Suspension | government |
| يارب _ارفع _عنا _البلاء | Oh Lord, raise us from calamity | Showing social |
| المسافة _ما _تفرقنا | distance does not separates us | solidarity |
| الحجر _المنزلي _واجب _وطني | Quarantine is a national duty | Supporting decisions |
| نبيها _صفر | We want it zero | taken by the government |

Table 1: Sample of hashtags used in this paper and the type of the hashtag used and populated by whom

| Hashtag Topics | Number of Tweets |
|---|---|
| Populated by Saudi governmental accounts | 851,095 |
| Discussing precautionary measures applied by the gov | 445,019 |
| Showing social solidarity | 177,412 |
| Supporting decisions taken by the government | 1,538,179 |

Table 2: Data set hashtag topics with the number of tweets for each topic

stead of taking tweets randomly, we wanted to use tweets from a normal Twitter user to be meaningful as sample to train our sentiment model, so we measured the average and try to take tweets that are around it since this is the number of characters most people tweets. As mentioned before, the average characters before cleaning was 184.63, and 142.23 characters after it. Then, we used the sample between these two numbers in-order to get tweets with an average length of characters to identify the sentiment. The total number of tweets in the sample is 129,391 tweets. By comparing the lexicon words to this dataset we were able to form a sentiment value for each tweet. The resulted labels are show in table 4 below.

The way each tweet was annotated was by checking each word in the tweet and comparing it with the lexicon data set we have. The lexicon data set labeled each word either positive or negative. We changed that to 1 and -1. When we went through each word in the tweet, we checked if it positive then we add 1 to the tweet score, if it is negative we subtract 1, and if the word is not in the lexicon

data set skip it. Finally if the tweet score is larger than zero we give it positive label, if it is less than zero we give it negative label, and if it is equal to zero it will get neutral as a label.

A binary classification models were built to be able to identify the sentiment label in the rest of the dataset. To build the models, we use two off-the-shelf machine learning algorithms: Naive Bayes and Support Vector Machine and train classifiers using Stratified 10-fold cross-validation with a ratio of 20:80 for testing and training. The model was built using only uni grams as features. Table 5 below shows the classification results for each model.

## 5 Results

### 5.1 What is the overall Sentiment towards the pandemic?

To have a better understanding of the public opinion towards the pandemic, we studied the sentiment people expressed in social media from the beginning of the pandemic in January 2020 until Saudi Arabia announced the lockdown measure in April 2020. Figure 1 shows the distribution of sentiment across time periods.

The results of the Support Vector Machine model and the Naive Bayes model are shown in table 4, we chose the one with better results and applied it on the whole dataset. The results in figure 1 and figure 2 are from the Support Vector Machine model.

Figure 1 shows that most tweets were expressing a neutral sentiment, that might have happened because most of the tweets contained prayers that does not express negative nor positive emotions.

| | Description | Dates |
|---|---|---|
| 1 | The date first case of COVID-19 identified in Wuhan until the disease was named COVID-19 by WHO | 31/12/19-11/02/20 |
| 2 | The date were the first COVID-19 confirmed case in KSA until KSA visits Suspension to GCC citizens | 02/02/20-28/02/20 |
| 3 | The date where arrival from other foreign countries need to be declared until WHO characterised COVID-19 as a pandemic | 03/03/20-11/03/20 |
| 4 | The date WHO characterized COVID-19 as a pandemic to the final date available in the dataset | 11/03/20-11/04/20 |

Table 3: Time periods used to divide the data set with the corresponding dates

| Label | Number of Tweets |
|---|---|
| Neutral | 66,141 |
| Positive | 34,410 |
| Negative | 28,840 |

Table 4: Sample data with sentiment labels

| Classifier | P | R | F-score | Acc. |
|---|---|---|---|---|
| NB | 87% | 84% | 86% | 87% |
| SVM | 98% | 97% | 98% | 98% |

Table 5: Classifier Results

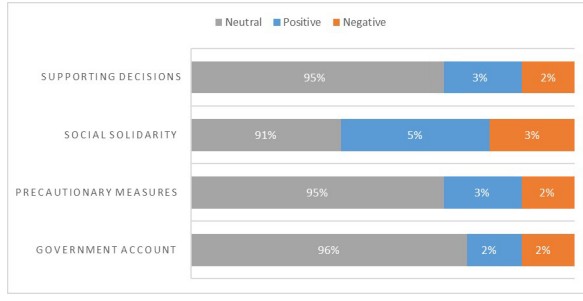

Figure 2: Sentiment distribution over topics

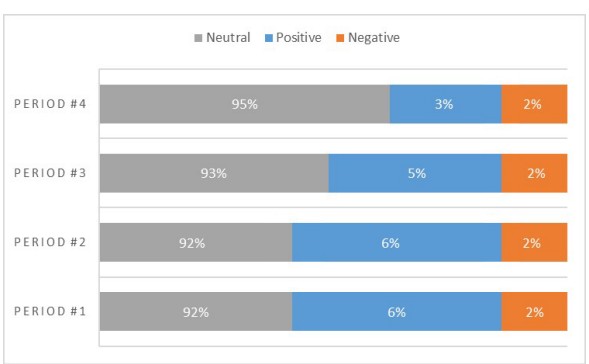

Figure 1: Sentiment distribution over the four time periods

Even though the sentiment distribution between time periods is very close, we can see that the first two periods had more positive sentiment compared to later periods. This might have happened because people might had high hopes that this virus is not staying for long and its temporary, which was proven wrong after a while. People expressed less positive emotions in the last period, that might be because the country started implementing precautionary measures as the lockdown which started on March 23th,2020. Moreover, there was not much of change in the expression of negative feelings even though the situation was escalating. That might

because the country implemented a good strategy in reassuring peoples concerns through spreading awareness materials and information through social media and other outlets. Also, the high trust the public have towards their governments and the validity of these precautionary measures.

### 5.2 How people reacted toward different topics?

To measure the public reactions towards this pandemic we used the hashtags categorization in the dataset. The topics where either showing social solidarity, discussing precautionary measures, supporting decisions taken by the government or using hashtags populated by the government to raise awareness and spread more information about the virus. The topics were labeled manually and based on the meaning and purpose of the hashtag. Table 1 shows a sample of the hashtags and their topics. Figure 2 shows the distribution of sentiment across different topics. The results show that people expressed more positive sentiment when they are supporting each other through the pandemic. These includes hashtags as, (translated from Arabic) "Lets Separate for our health", "distance does not separates us", "Oh Lord, raise us from calamity" and "Our merchants are good".

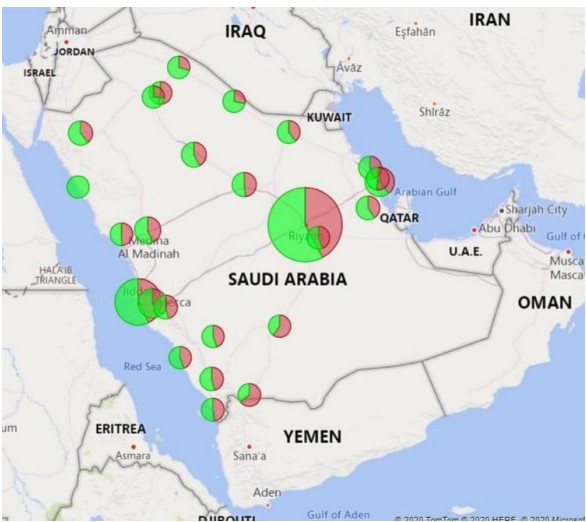

Figure 3: Distribution of sentiment across cites in Saudi Arabia

### 5.3 What is the geographical distribution of sentiment?

The geography of Saudi Arabia is big, with a 830K square miles. The cities of the country are spread across with huge distance in between its cities. Out of the total dataset only 42,238 had a location. of these tweets, 27,204 originated from Saudi Arabia. Figure 3 shows the amount of tweets and the distribution of sentiment across different cities. We can see that there is a high amount of tweets populated from the center, which is the Riyadh city, the capital city with 5,444 tweets expressing negative sentiment and 7,507 expressing positive sentiment. The second city is Jeddah with 2,570 tweets expressing negative sentiment and 7,507 expressing positive sentiment. Third comes Dammam city with 692 tweets expressing negative sentiment and 1,048 expressing positive sentiment. One of the highest city's in expressing negative emotions is Al-Qatif governorates, where the first case of COVID-19 has discovered.

## 6 Conclusion

In this research, our goal was to get a better understanding about Saudis' reactions towards Coronavirus. To do so, we explored the sentiment shared by the public with a focus on the analysis of Saudi dialect. Such understanding can contribute to inform fast-response policy making at early stages of a crisis. To achieve this goal, we built our own lexicon with 7,534 sentiment words, 1,808 of it was labeled as positive and 5,726 was labeled as negative. After that, we ran the lexicon on a sub-

set of the data. Then, two machine learning algorithms were used: NB and SVM with 10-fold cross-validation and ratio of 20:80 for both training and testing. The overall accuracy of classifiers were good, the SVM classifier was higher than NB with 98%, while the accuracy of NB was 87%. Our results showed that people had more optimistic sentiment and a high support for both government's decisions and precautionary measures to overcome the virus. However, one of the major limitations is the lack of resources to be able to analyze Arabic text, especially that there are different Arabic delicate for each country. As for future work we plan to understand the correlation between the different time periods and sentiment, we plan to get access to an extended version of the data set to continue the analysis. Moreover, the study of expressed emotions as fear and happiness is another direction to take such research.

## Acknowledgements

The authors gratefully thanks Ahmad Almutawa and Afnan Aloqaily and Rawan Almohimeed for their help in making this project happening.

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
