# OpenReview forum: "Tracking And Understanding Public Reaction During COVID-19: Saudi Arabia As A Use Case"
_EMNLP/2020/Workshop/NLP-COVID — NLP-COVID19-EMNLP Poster_

### Official Review · AnonReviewer1 · 2020-09-20
**Needs refinement**

**Rating:** 4
**Confidence:** 4

**Review:**

Overall: a good attempt to show NLP-related work on COVID-19 for Arabic language data.

However, the paper in its current form falls short of a few requirements, the most important of which is the lack of statistical testing and thereby the lack of alignment between the findings and the conclusions that are drawn.

Some comments:

- the introduction could do with some references of relevant work on social media studies around COVID-19 (incl. those of NLP workshops)
- line 83-84: needs a reference
- line 114: reference missing --> "?"
- line 116-120: the research questions are not motivated enough. Why is it interesting/necessary/worthwhile to look at sentiment? Why is it important to know the geographical distribution?
- section 2.2: this section seems redundant and could go I think.
- line 259: why do the tweets need to be related to influenza? This needs clarification.
- line 296-299: how were the keywords and hashtags categorised? Who did this? What was the annotators' agreement?
- line 317: the section numbering is off
- line 322: provide stats for the exclusions and general corpus descriptives
- line 353: what do we know about the validity of Crimson Hexagon?
- line 369-377: why were only tweets between 160-170 characters selected for sentiment identification?
- related: how was the labelling done? Provide details of the procedure.
- table 4: provide more detailed performance metrics
- fig 1: the figure is off - the percentages do not align when they should.
- line 440-470: this needs proper statistical testing - right now there is no way to assess the findings in a quantitative manner according to statistical standards
- line 476-478: how were the topics labelled?
- fig 2: are the differences really meaningful? Statistical testing would help understand this.
- fig 3: not really clear what the figure means and the use of spatial statistical models would be more appropriate here.
- lastly: at a few places, the wording is off.

---

### Official Review · AnonReviewer3 · 2020-09-23
**Tracking And Understanding Public Reaction During COVID-19: Saudi Arabia As A Use Case**

**Rating:** 4
**Confidence:** 4

**Review:**

**Summary**
This work described the sentiment analysis for Saudi Arabia based on the Twitter data from January 1, 2020, to April 10, 2020. The Twitter dataset was processed by Crimson Hexagon to identify both hashtag topics and geolocations of the posts. To identify the sentiment of the posts, sentiment lexicons were expanded based on the Arabic sentiment lexicons dataset by 4 annotators. To generate the ground truth sentiment dataset, more than 129K tweets were samples and annotated according to the expanded lexicons. NB and SVM models were trained and tested on the ground truth data. Based on the results, both sentiment distributions over time and topics were discovered. It also showed sentiment variations in different cities.

In this reviewer's view, this work is well structured. The 'Method' section needs more explanations and improvements.

**comments**
1. To expand the lexicon, 4 annotators extracted words expression emotions. After annotation, do they reach a certain agreement (Kappa statistics)?

2. Can you please clarify how to annotate the sentiment of posts based on the lexicons? Is human involved in this? If not, what is the algorithm to identify the sentiment of posts based on lexicons? One post may contain more than one sentiment keywords.

3. After training the SVM model (the best model shown in Table 4), please clarify if you apply this model to the entire dataset for the sentiment distribution in Figures 1 and 2.

---

### Official Review · AnonReviewer2 · 2020-09-27
**This is a worthwhile effort, but needs a bit more work**

**Rating:** 4
**Confidence:** 4

**Review:**

# [REVIEW] Tracking And Understanding Public Reaction During COVID-19: Saudi Arabia As A Use Case

EMNLP COVID19 — 26th Sep 2020


## SUMMARY

This paper seeks to use Twitter to better understand the emotional responses of the Saudi population to COVID-19.  As a first step, the researchers built an Arabic sentiment lexicon from several sources and augmented by some manually analysis to identify sentiment terms specific to the Saudi dialect.  Second, they geolocated the tweets using CrimsonHexagon.  Third, they assigned sentiment scores to tweets based on a process of (a) filtering tweets using the lexicon developed in step 1; and (b) applied NB & SVM to an annotated subset of these tweets [Note that this step is somewhat unclear].

Several key findings are presented.  First, public sentiment fluctuated substantially over the course of the pandemic, with sentiment improving (i.e. more positive tweets) over time.  Second, hashtags were used to home in on specific themes (e.g. support for decisions, social solidarity) should general support for government COVID-19 containment policies.  Third, the method was able to identify specific geographical locations that exhibited in their tweets greater than average negativity.

My overall assessment is that this is a worthwhile effort, but as it stands, the work is somewhat undercooked.  The methodology is not entirely clear.


## COMMENTS

1.  The writing can be a little hard to follow at points and generally would benefit from tightening up.  If we take the abstract as an example, there are a number of issues present:
	* “The COVID-19 had a great impact”
	* “While governments are taking extreme measures to **compact** the spread of the virus…”  I don’t think that “compact” is the right word here.
2.  The background for the work is extensive, but somewhat unfocussed.
3.  Some of the citations (e.g. Drus and Khalid, 2019) should probably be without brackets (i.e. \citet{} if you’re using latex)
4.  Missing citation on p2
5.  It sounds like the geolocation from CrimsonHexagon is something of a black box, given the quotation you use.  Is there any evidence regarding the reliability of their geolocation algorithms?
6.  Geographical distribution (Fig 3) looks about right given population centres in Saudi.